# Environmental Factors and Endometriosis

**DOI:** 10.3390/ijerph182111025

**Published:** 2021-10-20

**Authors:** Grzegorz Polak, Beata Banaszewska, Michał Filip, Michał Radwan, Artur Wdowiak

**Affiliations:** 1First Department of Gynaecological Oncology and Gynaecology, Medical University of Lublin, 20-081 Lublin, Poland; g.polak@umlub.pl; 2Department of Gynaecology, Obstetrics and Gynaecological Oncology, Poznan University of Medical Sciences, 60-533 Poznan, Poland; banaszewskabeata@ump.edu.pl; 3Department of Obstetrics and Pathology of Pregnancy, Medical University of Lublin, 20-081 Lublin, Poland; michalfilip@umlub.pl; 4Department of Gynaecology and Reproduction, “Gameta” Hospital, 95-030 Rzgow, Poland; mradwan@gameta.pl; 5Faculty of Health Sciences, Mazovian State University in Plock, 09-402 Plock, Poland; 6Chair of Obstetrics and Gynaecology, Faculty of Health Sciences, Medical University of Lublin, 20-081 Lublin, Poland

**Keywords:** endometriosis, environmental factors, diet, endocrine-disrupting chemicals

## Abstract

Endometriosis is a common disease, affecting up to 60–80% of women, with pelvic pain or/and infertility. Despite years of studies, its pathogenesis still remains enigmatic. Genetic, hormonal, environmental, and lifestyle-related factors may be involved in its pathogenesis. Thus, the design of the review was to discuss the possible role of environmental factors in the development of endometriosis. The results of individual studies greatly differ, making it very difficult to draw any definite conclusions. There is no reasonable consistency in the role of environmental factors in endometriosis etiopathogenesis.

## 1. Introduction

Endometriosis is a condition that occurs in women of childbearing potential, although it is only occasionally seen in prepubertal girls and postmenopausal women. It is defined as the presence of endometrial glands and stroma outside the uterine cavity. The prevalence of endometriosis reported in publications varies greatly, ranging from 5% to as much as 50% [1,2,3,4]. This wide variation may have several reasons. Not all cases of endometriosis are detected: in some women, where endometriosis is clinically asymptomatic, the condition can go undiagnosed or is incidentally discovered during surgical procedures performed for other reasons. Subtle endometrial lesions may go unnoticed even during surgery. On the other hand, the condition is more thoroughly looked for in patients undergoing surgery for infertility or chronic pelvic pain [1,2,3]. Endometriosis is estimated to be present in 2–22% of women without any clinical manifestations of the condition, 40–80% of patients with lower abdominal pain, and 13–48% of women with infertility [4]. Despite the discrepancies in the evaluation of its prevalence, endometriosis is undoubtedly one of the most commonly diagnosed and treated disorders of the female reproductive system.

Of the numerous theories attempting to explain the underlying mechanisms that lead to the development of endometriosis, several merit closer attention and may be grouped in three categories: in-situ theory, implantation theories, and induction theories. According to the in-situ theory, ectopic endometrium develops de novo from the multipotential peritoneal epithelium, from the germinal epithelium of the ovary, or from remnants of the Wolffian or Mullerian ducts. Implantation theories are based on the premise that the menstrual debris contains viable endometrial cells that can migrate, become implanted, and grow. Dissemination of endometrial cells may occur by extension, through the lymphatic vessels or blood vessels, through the ovaries during “retrograde menstruation”, or it may be iatrogenic and take place during surgical procedures. A hypothesis that combines these two groups of theories is the induction theory proposed by Levander. Based on his studies, Levander suggested that the exfoliated endometrial cells that arrive in the peritoneal cavity secrete active substances that stimulate the peritoneal epithelium to transform into endometrial tissue [4,5,6,7]. The aetiology of endometriosis remains, however, unclear and seems multifactorial. Genetic, hormonal, environmental, and lifestyle-related factors are involved in its pathogenesis [8].

## 2. Objective and Review Questions

Review on the literature suggests that there is no reasonable consistency in the role of environmental factors in endometriosis etiopathogenesis. The results of the trials presented by different authors are incoherent, and the conclusions are often contradictory. Thus, the design of the review was to discuss the possible role of ecological factors in the development of endometriosis. The aim was to collect the most recent data on environmental factors involved in the pathogenesis of the disease. We searched the MEDLINE and Embase databases to May 2021. The studies included were published during the last decade. A combination of the following medical subject headings or text words were used: endometriosis, dioxins, polychlorinated biphenyls, estrogens, diet, endocrine-disrupting chemicals, bisphenol A, and environmental factors. Endometriosis is a disease that, despite many years of research, is still surprising. Scientific papers describing its course are often contradictory and usually do not provide a consistent answer to puzzling issues. Due to the lack of uniform standards that adequately describe the disease, it is not possible to group scientific papers describing the topic.

## 3. Results and Discussion

### 3.1. In-Utero Exposure

Intrauterine exposure of female fetuses to certain substances is associated with a higher risk of endometriosis. Endometriosis is an estrogen-dependent disorder, and the ectopic endometrium, similarly to the normal uterine mucosa, expresses estrogen receptors [9]. There are currently no substantiated scientific data to demonstrate the benefits of estradiol to human fetuses. Administration of ethinyl estradiol, a common ingredient of oral contraceptives, to pregnant mice results in an increased incidence of adenomyosis, or endometriosis of the uterine muscle [10]. Ethinyl estradiol is a hormone that falls into category X, which means that studies in animals or humans have demonstrated fetal abnormalities and/or there is positive evidence of human fetal risk based on adverse reaction data from investigational or marketing experience, and the risks involved in use of the drug in pregnant women clearly outweigh potential benefits. Diethylstilbesterol (DES) is a drug that was used to prevent premature labor until 1971, when it was confirmed to exert teratogenic effects [11]. Analysis of the Nurses’ Health Study unequivocally demonstrated an increased risk of laparoscopically confirmed endometriosis in women who had been exposed to DES as fetuses. Interestingly, women who were born of twin pregnancies had additionally increased risk of endometriosis, which, hypothetically, could have been due to the higher estrogen concentrations that typically accompany such pregnancies [12]. The association between in-utero exposure to DES and the risk of endometriosis has also been confirmed in other studies [13]. The exact mechanism responsible for this association remains unknown. Golden et al. demonstrated that in utero exposure to DES interferes with the expression of estrogen receptors [14]. Exposure of mice to DES results in abnormal expression of estrogen receptor-dependent genes that encode lactoferrin and epidermal growth factor in the uterus and vagina [10]. Exposure to DES during pregnancy also leads to structural abnormalities of the uterus in female fetuses. The classification of congenital anomalies of the uterus includes a separate category of “developmental anomalies associated with diethylstilbesterol use” [15]. Congenital anomalies of the uterus, by augmenting “retrograde menstruation”, are a recognized risk factor for endometriosis [5].

Exposure of pregnant women to toxins can also be associated with a higher risk of endometriosis in their daughters. This association is particularly well understood in cases of exposure to dioxins and polychlorinated biphenyls. A study in an animal model demonstrated that prenatal exposure to 2,3,7,8-tetrachlorodibenzodioxin (TCDD) increased the risk of surgically induced endometriosis in adult mice and rats [16]. This phenomenon may be associated with progesterone resistance [17]. It was hypothesized that exposure to TCDD and other dioxins in fetal life could induce endometriosis as a result of epigenetic modifications, interfering with the expression of genes involved in the pathogenesis of this condition [18]. In-utero exposure of mice to bisphenol A (BPA) is also associated with the risk of endometriosis in later life [19]. This effect is most likely due to the unbalanced hyperestrogenism caused by the inhibited expression of progesterone receptors on the one hand and the increased estrogen concentration on the other [20].

It is estimated that approximately 1.7% pregnant women worldwide smoke [21]. Smoking during pregnancy negatively impacts embryofetal growth and development. It increases the risk of miscarriage, premature labor, intrauterine growth restriction, and many other complications of pregnancy [22,23]. Smoking has, however, been shown to reduce the risk of endometriosis in later life among female fetuses [24]. The underlying mechanism is unknown. Nicotine, along with its metabolite cotinine, may suppress aromatase-dependent conversion of androgens to estrogen, stimulate apoptosis, and inhibit angiogenesis [20]. Both hyperestrogenism and neovascularization play a fundamental role in the development of this disorder, which is why nicotine may inhibit the development of endometriosis (see Table 1).

### 3.2. Diet

Results of many studies point to an association between diet and the development of endometriosis. This association is, however, unclear. It may be speculated that dietary factors affect the formation of sex hormones and exert pro- or antioxidant effects and proinflammatory effects. All these factors play a considerable role in the pathogenesis of endometriosis. At the same time, the effect of contaminants accompanying food production on the development of this condition cannot be ruled out. In 2004, it was shown that women with laparoscopically confirmed endometriosis consumed smaller quantities of fresh fruits and green vegetables compared to women without endometriosis [25]. Harris et al. also demonstrated that consumption of fresh fruits, including, in particular, citrus fruits, reduced the risk of this disorder [26]. A diet rich in fruits provides large amounts of provitamin A, lower levels of which have been reported in patients with endometriosis [27]. Vitamin A suppresses the formation of the proinflammatory interleukin-6, a cytokine whose high levels are found in the amniotic fluid in women with endometriosis [28,29]. Citrus fruits also contain high quantities of vitamin C, which inhibits inflammation and exerts antioxidant effects. Vegetables and particularly fruits may contain organochlorines, which in turn have been positively associated with the risk of endometriosis [30,31]. Paradoxically, Trabert et al. showed an increased risk of endometriosis in American women consuming large amounts of fruits [32]. Hypothetically, this may be associated with the large quantity of pesticides used during cultivation in the United States.

A diet rich in red meat increases the risk of endometriosis. The Nurses’ Health Study II, which included nearly 82,000 American nurses, demonstrated over 22 years of follow-up that consumption of both processed and unprocessed red meats increased the risk of endometriosis. This association was strongest among women who had never reported infertility and thus were more likely to present with pain symptoms. On the other hand, intakes of poultry, fish, shellfish, and eggs were shown to be unrelated to endometriosis risk [33]. This study confirmed the findings of Parazzini et al., who also demonstrated an association between the consumption of red meat and the risk of endometriosis [25]. This effect may be due to the pro-oxidative effects of haem released from red meat. Haem is involved in the initiation of lipid peroxidation, in which unsaturated fatty acids or other lipids are oxidated by free radicals to their respective peroxides [34]. Disturbances of the pro-/antioxidant balance that lead to oxidative stress are a recognized mechanism responsible for the development of endometriosis [35]. A diet rich in red meat can also be responsible for higher estradiol levels by which it leads to endometriosis. Harmon et al. demonstrated that vegetarian women have lower serum levels of estrone and estradiol compared to women who consume large amounts of red meat [36]. Palmitic acid was the only saturated fatty acid that was significantly associated with increased risk of endometriosis. High intake of trans-unsaturated fats was also a risk factor for endometriosis, whereas intake of long-chain omega-3 fat acids decreased the risk of endometriosis (see Table 2) [37].

### 3.3. Endocrine-Disrupting Chemicals

Endometriosis is an estrogen-dependent disorder that develops in women of childbearing potential. Its pathogenesis is therefore affected by endocrine-disrupting chemicals (EDCs), which are defined as a group of exogenous chemical compounds that affect the function of the endocrine system. The endocrine-disrupting chemical is defined as “an exogenous chemical, or mixture of chemicals, that interferes with any aspect of hormone action” [38]. The continuously updated list of EDCs maintained by TEDX (The Endocrine Disruption Exchange) has grown to include almost 1500 substances at the moment [39]. Endocrine-disrupting chemicals enter the human body with food, water, dust by inhalation, with the inspired air, and via the transdermal route after using cosmetics and creams. Transplacental transfer of these substances to the developing fetus has also been demonstrated [40]. They are also found in human milk [40]. EDCs accumulate in the adipose tissue, and the effects of their metabolites persist for a long time [41]. Because of the potential association between exposure to EDCs and the development of endometriosis, many studies have been devoted to this topic. Such studies are difficult to design, as it is difficult to identify both the study group and the control group and to measure the exposure to EDCs and the effects of other factors on the development of this condition. An additional difficulty is associated with the fact that endometriosis may take several different forms (ovarian endometrioma, peritoneal endometriosis, deeply infiltrating endometriosis, and adenomyosis—or endometriosis of the uterine muscle), which not only differ in location but also have different clinical presentations. In some cases, endometriosis remains asymptomatic, and a certain diagnosis can only be established by invasive evaluation and histopathological confirmation. Silent endometriosis is a condition in which the patient does not experience any discomfort resulting from the development of the disease. Symptoms may appear in later life or remain dormant.

Cosmetics and personal care products (PCPs) contain numerous EDCs [42]. They are increasingly used worldwide. In the USA alone, the cosmetics and PCPs market is estimated at €78.6 billion [43]. EDCs released by cosmetics and PCPs are mainly associated with the benzophenone (BP) and paraben (PB) families. BPs include BP-1, BP-3, and 4-hydroxybenzophenone (4-OH-BP) and are used as UV blockers in suntanning creams and other cosmetics [44]. Frequent use of cosmetics and PCPs results in higher urinary concentrations of these substances and may potentially lead to endometriosis. This is likely associated with the estrogenic and pro-oxidative effects of these compounds [45]. Oxidative stress is a recognized factor involved in the pathogenesis of endometriosis [34]. By damaging germinal cells and disrupting folliculogenesis, EDCs adversely affect fertility, which further exacerbates the problem of infertility in women with endometriosis [46].

Of the many EDCs, compounds that are best understood in terms of potential involvement in the pathogenesis of endometriosis are polychlorinated biphenyls (PCBs) (congeners 118, 138, 153, and 156) and dioxins (2,3,7,8-TCDD, 1,2,3,7,8-PeCDD, 1,2,3,4,7,8-HxCDD, and 1,2,3,6,7,8-HxCDD).

Bisphenol A (BPA) disrupts the pulsatile secretion of GnRH, which negatively affects the function of the hypothalamic-pituitary-ovarian axis. Pre- and postnatal exposure to BPA induces future structural changes in the ovaries, which further contributes to abnormal folliculogenesis. It may also adversely affect the structure of the uterus [40], and by increasing retrograde menstruation, it may increase the risk of endometriosis. BPA is able to exerts its endocrine effects, as it displays affinity for the estrogen receptors Erα and Erβ [47]. A review article by Anger and Foster suggests that BPA, despite its high environmental distribution, is not the leading cause of endometriosis. The main reasons are the short half-life in the human body and too-low doses to which the studied women were exposed. Researchers suggest that unequivocal results may be due to the long time lag between diagnosis and measurement for exposure [48]. BPA, phthalates, and perfluoroalkyl substances (PFAS) present in water and foods adversely affect puberty in teenagers, and in addition to being involved in the pathogenesis of endometriosis, they increase the risk of polycystic ovarian syndrome (PCOS) and recurrent pregnancy in humans and animals [49]. Studies in mice have demonstrated that exposure to BPA induces endometriotic foci in the adipose tissue that surround the reproductive organs in these animals. BPA has also been shown to induce cystic ovaries, adenomatous hyperplasia with cystic endometrial hyperplasia, and atypical hyperplasia [18]. Results of studies investigating the potential involvement of BPA in the pathogenesis of endometriosis are conflicting. Most studies have, however, reported higher urinary levels of BPA in women with endometriosis compared to those without it [50,51,52]. Higher urinary levels of BPA may only be associated with the peritoneal form of the condition and may not be present in endometriosis located elsewhere [53].

Dioxins are considered to be the most toxic industrial pollutants that are highly resistant to degradation and, due to their lipophilic nature, show considerable bioaccumulation [54]. The studies showing the influence of dioxins on the occurrence and development of endometriosis are not unequivocal. In Rier SE et al., the increase in the concentration of 2,3,7,8-TCDD was significantly correlated with endometriosis; however, one should remember about the differences between the organism of the rhesus monkey and that of humans. In the human organism, the adipose tissue accumulating 2,3,7,8-TCDD is significantly better developed, which means that the accumulation effect may be less intense [55]. Studies in women have generated conflicting results on the association between the exposure to 2,3,7,8-TCDD and the development of endometriosis. Mayani et al. found higher blood levels of 2,3,7,8-TCDD in women with endometriosis compared to those without this condition while demonstrating no correlation between the levels of this dioxin and the severity of endometriosis [56]. In 2002, results were published of a study evaluating the risk of endometriosis 20 years after the chemical plant explosion near Seveso, Italy, that resulted in the highest populational exposure to 2,3,7,8-TCDD ever reported. The authors found twice as high but statistically non-significant risk of endometriosis in women whose blood samples collected after the accident displayed 2,3,7,8-TCDD concentrations exceeding 100 ppt [57]. An association has also been shown between increased dioxin accumulation in the adipose tissue and the risk of deeply infiltrating endometriosis [58]. It has been suggested in recent years that the development of endometriosis may be associated with epigenetic alterations [59]. In this mechanism, dioxins may be responsible for the pathogenesis of this condition, as they affect histone modification, DNA methylation, and expression of non-coding RNA (ncRNA) [18]. A study in an animal model has shown that the suppression of the epigenetic histone modification caused by 2,3,7,8-TCDD delays the progression of endometriosis [60]. However, not all the studies have confirmed the association between the exposure to dioxins and the risk of endometriosis [61,62,63,64]. Epigenetic changes occurring in the genetic material modulate susceptibility to endometriosis. Research varies in terms of impact, as epigenetic changes are not constant and are constantly changing. This is due to the regular change in environmental factors [65].

Endometriosis is a condition of multifactorial etiology. Despite years of studies, its pathogenesis has not been fully elucidated. Gaining an understanding of its underlying mechanisms would enable effective prevention and treatment of this condition. Genetic factors play a huge role in the pathogenesis of endometriosis. The risk of endometriosis is 6–9% higher in first-degree relatives of women with endometriosis and 15% higher when they had severe disease [59]. The genes responsible for the heritability of the disease have not, however, been identified. The involvement of epigenetic factors in their modification cannot be ruled out. Such effects may be exerted by environmental findings and, by controlling gene activity, may be involved in the pathogenesis of endometriosis (see Table 3).

## 4. Conclusions

Designing studies to investigate the potential involvement of environmental factors in the pathogenesis of endometriosis is extremely difficult. Endometriosis is a heterogenous disorder that occurs in various forms and locations and has varied symptomatology. In some cases, it is asymptomatic; for a definite diagnosis to be established, histopathological examination needs to be carried out, which in turn requires an invasive approach. Other difficulties are associated with the selection of an appropriate control group and the exact evaluation of the dose and duration of exposure to the substance of interest. The time elapsed from the exposure to the substance of interest to the development of endometriosis induced by it is unknown. It is also impossible to determine the exact onset of the condition. In addition, due to the polyetiological and still unclear nature of endometriosis, it is impossible to eliminate the impact of other factors involved in its pathogenesis, which is why the results of individual studies may greatly differ, making it very difficult to draw any definite conclusions.

## Figures and Tables

**Table 1 ijerph-18-11025-t001:** In-utero factors of endometriosis.

Author	Year of Publishing	Factor	Effect on Endometriosis
Koike E, Yasuda Y, Shiota M, et al. [10]	2013	Ethynil estradiol	Increased incidence of adenomyosis or endometriosis
Herbst AL, Ulfelder H, Poskanzer DC, Longo LD [11]	1999	Diethylstilbesterol (DES)	Increased risk of laparoscopically confirmed endometriosis
Missmer SA, Hankinson SE, Spiegelman D, et al. [12]	2004	High estrogen level due to twin pregnancy	Increased risk of endometriosis
Benagiano G, Brosens, I. [13]	2014	Diethylstilbesterol (DES)	Increased risk of endometriosis
Golden RJ, Noller KL, Titus-Ernstoff L, et al. [14]	1998	Diethylstilbesterol (DES)	High expression of estrogen receptors
Buck Louis GM, Hediger ML, Peña JB. [24]	2007	Smoking (nicotine)	Reduce the risk of endometriosis

**Table 2 ijerph-18-11025-t002:** Diet factors of endometriosis.

Author	Year of Publishing	Factor	Effect on Endometriosis
Parazzini F, Chiaffarino F, Surace M, et al. [25]	2004	Consumed smaller quantities of fresh fruits and green vegetables	Increased risk of endometriosis
Harris HR, Eke AC, Chavarro JE, Missmer SA. [26]	2018	Consumption of fresh fruits	Reduced the risk of endometriosis
Mier-Cabrera J, Aburto-Soto T, Burrola-Mendez S, et al. [27]	2009	Low level of vitamin A	Increased risk of endometriosis
Grassi P, Fattore E, Generoso C, et al. [30]	2010	Organochlorines	Increased risk of endometriosis
Buck Louis GM, Chen Z, Peterson CM, et al. [31]	2012	Organochlorines	Increased risk of endometriosis
Trabert B, Peters U, De Roos AJ, et al. [32]	2010	Large amounts of fruits	Increased risk of endometriosis
Yamamoto A, Harris HR, Vitonis AF, et al. [33]	2018	Red-meat-rich diet	Increased risk of endometriosis
Parazzini F, Chiaffarino F, Surace M, et al. [25]	2004	Red-meat-rich diet	Increased risk of endometriosis
Yamamoto A, Harris HR, Vitonis AF, et al. [33]	2018	Poultry-, fish-, shellfish-, and eggs-rich diet	Unrelated to endometriosis risk
Missmer S, Chavarro JE, Malspeis S, et al. [37]	2010	Palmitic acid/ trans-unsaturated fats	Increased risk of endometriosis
Missmer S, Chavarro JE, Malspeis S, et al. [37]	2010	Long-chain omega-3 fat acids	Reduce the risk of endometriosis

**Table 3 ijerph-18-11025-t003:** EDC factors of endometriosis.

Author	Year of Publishing	Factor	Effect on Endometriosis
Peinado FM, Ocón-Hernández O, Iribarne-Durán LM, et al. [45]	2020	Cosmetics and personal care products (PCPs)	Potentially lead to endometriosis
Anger DL, Foster WG. [48]	2008	Bisphenol A (BPA)	No effect on the development of endometriosis
Upson K, Sathyanarayana S, De Roos AJ, et al. [53]	2014	High urinary level of BPA	Associacion with the peritoneal form of endometriosis
Cobellis L, Colacurci N, Trabucco E, et al. [50]	2009	High urinary level of BPA	Potentially lead to endometriosis
Simonelli A, Guadagni R, De Franciscis P, et al. [51]	2017	High urinary level of BPA	Potentially lead to endometriosis
Rashidi BH, Amanlou M, Lak TB, et al. [52]	2017	High urinary level of BPA	Potentially lead to endometriosis
Rier SE, Martin DC, Bowman RE, et al. [55]	1993	2,3,7,8-TCDD	Significantly correlation with endometriosis
Mayani A, Barel S, Soback S, Almagor, M. [56]	1997	2,3,7,8-TCDD	No correlation with endometriosis
Eskenazi B, Mocarelli P, Warner M, et al. [57]	2002	2,3,7,8-TCDD	High but statistically non-significant risk of endometriosis
Martınez-Zamora MA, Mattioli L, Parera J, et al. [58]	2015	2,3,7,8-TCDD	Associacion with deeply infiltrating endometriosis
Tsukino H, Hanaoka T, Sasaki H, et al. [64]	2005	Exposure to dioxins	No effect on the development of endometriosis
Lebel G, Dodin S, Ayotte P, et al. [61]	1998	Exposure to dioxins	No effect on the development of endometriosis
De Felip E, Porpora MG, di Domenico A, et al. [62]	2004	Exposure to dioxins	No effect on the development of endometriosis
Pauwels A, Schepens PJ, D’Hooghe T, et al. [63]	2001	Exposure to dioxins	No effect on the development of endometriosis

## Data Availability

The datasets generated during and/or analyzed during the current study are available from the corresponding author on reasonable request.

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
