# Peer review of "Environmental Factors and Endometriosis"

_ijerph, 2021, doi:10.3390/ijerph182111025_

Round 1

Reviewer 1 Report

This review by Polak et al entitled “Environmental factors and endometriosis” is a reasonable update on factors contributing to endometriosis. Many of the studies cited are from the past 5 years. This review is relatively superficial, and could benefit from a more in-depth examination of potential molecular mechanisms at play with each compound discussed. Aside from that suggestion, I have relatively minor comments.

Line 26- please cite which studies put endometriosis prevalence at 50% (of total individuals of childbearing potential?), as this number seems unlikely.

Line 67- regarding the Koike et al study from 2013- while their data and experimental design look good, under what circumstances would a pregnant person be treated with ethinyl estradiol? I could find only one paper from 1952 on the potential use of EE2 during pregnancy (to prevent fetal demise), is this really a relevant compound to include in this review?

Line 69- please define “category X”

Line 142- “Disturbances of the pro-/antioxidant balance that lead to antioxidative stress are a recognised mechanism responsible for the development of endometriosis”. “Antioxidative stress” should be “oxidative stress”- antioxidant stress is defined as the overabundance of antioxidant compounds, while your 2018 paper showed that the peritoneal fluid of people with endometriosis had lower antioxidants (page 589- “PF Total Antioxidant Status was significantly (p<0.01) lower in women with endometriosis”).

Line 161- transplacental crossing of EDCs- citations needed.

Lines 167-171- this description of various forms of endometriosis should be in the introduction and elaborated on further.

Lines 190-211 (BPA paragraph)- you don’t mention endometriosis until the last three sentences of the paragraph, and at that point state that studies are conflicting regarding the role of BPA during endometriosis. Cut back the discussion of BPA causing non-endometriosis reproductive complications and instead elaborate on the studies showing both involvement and non-involvement of BPA in endometriosis.

Lines 233-234- discuss potential confounders of why studies differ on the effects of dioxin exposure on development of endometriosis.

Lines 235-243- move this paragraph to the discussion, it does not belong in the EDC section.

Author Response

Dear All

Thank you very much for review of our paper. We carefully studied the suggestions and raised questions and we improved our work according to the referees remarks.

Line 26: citation number [3] and [4]

Line 67: we added: There are currently no substantiated scientific data to demonstrate the benefits of estradiol to human fetuses.

Line 69: we added: Category X means that studies in animals or humans have demonstrated fetal abnormalities and/or there is positive evidence of human fetal risk based on adverse reaction data from investigational or marketing experience, and the risks involved in use of the drug in pregnant women clearly outweigh potential benefits.

Line 142: we changed antiooxidative stress into oxidative stress.

Line 161: citation number [40].

Line 167-171:  Included in the Introduction(line 26-29)- Not all cases of endometriosis are detected: in some women, where endometriosis is clinically asymptomatic, the condition can go undiagnosed or is incidentally discovered during surgical procedures performed for other reasons.

We added: Silent endometriosis is a condition in which the patient does not experience any discomfort resulting from the development of the disease. Symptoms may appear in later life or remain dormant.

Line 190-211(BPA): Bisphenol A (BPA) disrupts the pulsatile secretion of GnRH, which negatively affects the function of the hypothalamic-pituitary-ovarian axis. Pre- and postnatal exposure to BPA induces future structural changes in the ovaries, which further contributes to abnormal folliculogenesis. It may also adversely affect the structure of the uterus [40] and, by increasing retrograde menstruation, it may increase the risk of endometriosis. BPA is able to exerts its endocrine effects, as it displays affinity for the estrogen receptors Erα and Erβ [47]. BPA has also been shown to suppress the effects of androgens by blocking their receptors [48]. Even though up to about 90% of exposure to BPA occurs via food, it should still be borne in mind that this compound may also be found in baby bottles. It is released from food containers at high temperatures and at alkaline pH [49]. BPA, phthalates and perfluoroalkyl substances (PFAS) present in water and foods adversely affect puberty in teenagers and, in addition to being involved in the pathogenesis of endometriosis, they increase the risk of polycystic ovarian syndrome (PCOS) and recurrent pregnancy in humans and animals [50]. Studies in mice have demonstrated that exposure to BPA induces endometriotic foci in the adipose tissue that surround the reproductive organs in these animals. BPA has also been shown to induce cystic ovaries, adenomatous hyperplasia with cystic endometrial hyperplasia, and atypical hyperplasia [18].

We added: Review article by Anger and Foster suggests that BPA, despite its high environmental distribution, is not the leading cause of endometriosis. The main reasons are the short half-life in the human body and too low doses to which the studied women were exposed. Researchers suggest that unequivocal results may be due to the long time lag between diagnosis and measurement for exposure.

Line 233-234:  Dioxins are considered to be the most toxic industrial pollutants that are highly resistant to degradation and, due to their lipophilic nature, show considerable bioaccumulation [55].

We added: The studies showing the influence of dioxins on the occurrence and development of endometriosis are not unequivocal. In the Rier SE et all. the increase in the concentration of 2,3,7,8-TCDD was significantly correlated with endometriosis, however, one should remember about the differences between the organism of the rhesus monkey and that of humans. In the human organism, the adipose tissue accumulating 2,3,7,8-TCDD is significantly better developed, which means that the accumulation effect may be less intense. Already in 1993, it was shown that exposure of rhesus monkeys to 2,3,7,8-TCDD stimulated the development of endometriosis in a dose-dependent manner [56]. Studies in women have, however, generated conflicting results on the association between the exposure to 2,3,7,8-TCDD and the development of endometriosis. Mayani et al. found higher blood levels of 2,3,7,8-TCDD in women with endometriosis compared to those without this condition, while demonstrating no correlation between the levels of this dioxin and the severity of endometriosis [57].. In 2002, results were published of a study evaluating the risk of endometriosis 20 years after the chemical plant explosion near Seveso, Italy, that resulted in the highest populational exposure to 2,3,7,8-TCDD ever reported. The authors found twice as high but statistically non-significant risk of endometriosis in women whose blood samples collected after the accident displayed 2,3,7,8-TCDD concentrations exceeding 100 ppt [58]. An association has also been shown between increased dioxin accumulation in the adipose tissue and the risk of deeply infiltrating endometriosis [59]. It has been suggested in recent years that the development of endometriosis may be associated with epigenetic alterations [60]. In this mechanism, dioxins may be responsible for the pathogenesis of this condition, as they affect histone modification, DNA methylation, and expression of non-coding RNA (ncRNA) [18]. A study in an animal model has shown that the suppression of the epigenetic histone modification caused by 2,3,7,8-TCDD delays the progression of endometriosis [61]. However, not all the studies have confirmed the association between the exposure to dioxins and the risk of endometriosis [62–65].

We added: Epigenetic changes occurring in the genetic material modulate susceptibility to endometriosis. Research varies in terms of impact as epigenetic changes are not constant and are constantly changing. This is due to the regular change in environmental factors.

We hope that our revision will meet your expectations.

Please find enclosed the revised manuscript.

Yours sincerely

Prof. Grzegorz Polak, MD, PhD       

Reviewer 2 Report

See attached file

Author Response

Dear Reviewer

Thank you very much for review of our paper. We carefully studied the suggestions and raised questions and we improved our work according to the referees remarks.

We consider that the authors should present a detailed explanation of the selection methodology of the texts they have reviewed for the preparation of this work, selection criteria and exclusion criteria for articles from the literature. Likewise, detail those corresponding to each of the sections outlined in the results paragraph.

We added:

We searched the MEDLINE and Embase databases to May 2021. A combination of the following medical subject headings or text words were used: endometriosis, dioxins, polychlorinated biphenyls, estrogens, diet, endocrine-disrupting chemicals, bisphenol A, environmental factors.

We recommend that the authors present the results in a table with the most relevant points of their work. In the same way, they should also synthesize those areas of research or future lines of study to clarify the points of uncertainty in the genesis of the disease.

Thank you for the important tip. However, we believe that the problem presented in this article is extremely complicated. Various chemicals have been described that may be involved in the pathogenesis of endometriosis in different ways. For this reason, table data is not feasible and would distort the descriptive nature of the results.

We hope that our revision will meet your expectations.

Please find enclosed the revised manuscript.

Yours sincerely

Prof. Grzegorz Polak, MD, PhD        

Round 2

Reviewer 2 Report

Attached document

Author Response

Author response to the first round review:

Dear All

Thank you very much for review of our paper. We carefully studied the suggestions and raised questions and we improved our work according to the referees remarks.

Comments to authors The detailed analysis of this article that seems to us of special interest in a field as controversial as the genesis of endometriosis and its inducing factors. We consider that the authors should present a detailed explanation of the selection methodology of the texts they have reviewed for the preparation of this work, selection criteria and exclusion criteria for articles from the literature. 

We added:

Endometriosis is a disease that, despite many years of research, is still surprising. Scientific papers describing its course are often contradictory and usually do not provide a consistent answer to puzzling issues. Due to the lack of uniform standards that adequately describe the disease, it is not possible to group scientific papers describing the topic.

However, the criteria for selecting the exclusion were not unequivocally specified due to the too wide scope of the topic, which made it possible to describe the problem in a broader manner. All scientific articles that were considered come from recognized scientific journals.

We recommend that the authors present the results in a table with the most relevant points of their work. In the same way, they should also synthesize those areas of research or future lines of study to clarify the points of uncertainty in the genesis of the disease.

As you ask we added tables with the most relevant points. The issue is so obscure that the future area of ​​research cannot focus on one area.

We hope that our revision will meet your expectations.

Please find enclosed the revised manuscript.

Yours sincerely

Author response to the second round review:

Dear reviewers

Thank you very much for you letter. We believe that your suggestions may upgrade our work.

We must emphasize that our article is not a meta-analysis, which is extremely difficult to write due to the specificity of the problem, but a review work. We searched the MEDLINE and Embase databases to May 2021. A combination of the following medical subject headings or text words were used: endometriosis, dioxins, polychlorinated biphenyls, estrogens, diet, endocrine-disrupting chemicals, bisphenol A, environmental factors.

The eligibility criteria we us were not unequivocally specified due to the too wide scope of the topic, which made it possible to describe the problem in a broader manner. All scientific articles that were considered come from recognized scientific journals. All articles were published in English language.

We hope that our answer will meet your expectations.

Yours sincerely

Prof. Artur Wdowiak MD, PhD

Round 3

Reviewer 2 Report

attached
